# PHYSICS-AWARE CAUSAL GRAPH NETWORK FOR SPATIOTEMPORAL MODELING

## ABSTRACT

Interpretable physics equations are widely recognized as valuable inductive biases for constructing robust spatiotemporal models. To harness these valuable pieces of knowledge, existing approaches often presuppose access to the exact underlying equations. However, such an assumption usually doesn't hold, especially in the context of real-world observations. Conversely, causality systematically captures the fundamental causal relations across space and time that are intrinsically present in physics dynamics. Nevertheless, causality is often ignored as a means of integrating prior physics knowledge. In this work, we propose a novel approach that effectively captures and leverages causality to integrate physics equations into spatiotemporal models, without assuming access to precise physics principles. Specifically, we introduce a physics-aware spatiotemporal causal graph network (P-STCGN). Causal relationships are analytically derived from prior physics knowledge and serve as physics-aware causality labels. A causal module is introduced to learn causal weights from spatially close and temporally past observations to current observations via semi-supervised learning. Given the learned causal structure, a forecasting module is introduced to perform predictions guided by the cause-effect relations. Extensive experiments on time series data show that our semi-supervised causal learning approach is robust with noisy and limited data. Furthermore, our evaluations on real-world graph signals demonstrate superior forecasting performance, achieved by utilizing prior physics knowledge from a causal perspective.

## 1 INTRODUCTION

Spatiotemporal modeling has drawn significant interest recently due to its wide application in climate (Faghmous & Kumar, 2014), traffic systems (Ermagun & Levinson, 2018), electricity networks (Masi et al., 2009), and many other fields. Employing deep neural networks has demonstrated superior performance, particularly in data-rich settings. The spatiotemporal observations in the physical world, such as the climate and weather measurements (Kashinath et al., 2021), inherently follow physical principles. Hence, physics equations are usually recognized as valuable information for robust spatiotemporal modeling. Integrating domain knowledge with data-driven models has emerged as one of the most promising directions forward, clearing a path for the construction of more robust and interpretable pipelines.

Many excellent studies have been conducted on physics-informed machine learning (Wang et al., 2020a; Greydanus et al., 2019; Raissi et al., 2019). One successful approach to building physics-informed models is to assume access to precise physics principles. Consequently, these models incorporate the identified equations into deep models in a hard way (e.g., through architectural modifications (Wang et al., 2020b)). However, the assumption of the correctness of physics knowledge can be problematic, as these physics principles can be fragile or imperfect in real-world applications (Finzi et al., 2021; Wang et al., 2022). In spatiotemporal modeling, addressing such concerns becomes more important, as we frequently have access to simple equations that partially describe the dynamics we intend to model. However, the exact form of these equations and their associated physics parameters remains elusive. Hence, there arises a need for soft integration to bridge the gap between our known prior knowledge and the unknown principles governing real-world observations.

Causality is important for systematically harnessing the structured knowledge embedded in physics equations. This is grounded in the fundamental principle that *an effect can not occur before its cause*.

Observations from the same dynamic system, even when driven by different underlying physics equations, share common causal relations across space and time that intrinsically present in the dynamics. This insights motivates us to employ causal relations as a soft encoding of the underlying dynamics. Existing soft integration methods often rely on heuristic approaches, primarily fusing information at an intractable feature level (Finzi et al., 2021; Takeishi & Kalousis, 2021; Seo et al., 2021). The inference of causal relations from prior physical knowledge is rarely explored as a means of soft integration.

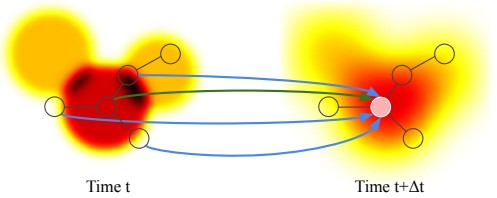

Time t          Time t+Δt

Figure 1: Heat dissipation over 2D space and time. Nodes in the graph structure correspond to sensors and the observations at each sensor are time varying. Given the heat equation ($\dot{u} = D\Delta u$), we can provide spatial (blue) and temporal (green) causal relations from previous nodes to a current target node (white).

In particular, given an equation that's moderately beneficial for understanding the target dynamics, we can decompose it into causes and effects analytically. For example, when the heat equation ($\frac{\partial u}{\partial t} = D\Delta u$ where $D$ is a diffusivity constant) is considered, we know that temporally first order and spatially second order derivatives are involved. We then specify causes and effects on a discrete domain (time interval $\Delta t$) as:

$$u_i(t+1) = u_i(t) + \Delta t \cdot D\Delta u, = u_i(t) + \Delta t \cdot D \sum_{j \in \mathcal{N}_i} (u_i(t) - u_j(t)), \qquad (1)$$

where $\Delta$ is the Laplace operator and $\mathcal{N}_i$ is a set of adjacent nodes of $i$-th node. Eq. 1 shows the discrete Laplace operator. For the target value $u_i(t+1)$, the variables in the right-hand side are regarded as *known* causes from the heat equation. To softly incorporate this physics equation without assuming its preciseness, we utilize it as a basis for extracting causal labels. These labels are employed for a semi-supervised causal structure learning, i.e., assigning explicit causal labels between the subset of nodes associated with the equation (Fig. 1). They are used for regularization such that the model can capture the causal structures that align with the physics-aware causality. Notably, unlike most causal discovery from data approaches, we employ physics-aware causality for semi-supervised causal structure learning. More detailed discussions on related works regarding physics-informed spatiotemporal modeling and causal discovery in time series can be found in the Appx. A.

In summary, we introduce a novel physics-aware causal graph network (P-STCGN) for the soft integration of physics laws. Specifically, in our modeling process, we decouple causal structure learning from dynamic forecasting. Physics-aware causality is derived from prior physics knowledge. A causal module is introduced to learn causal relations partially from analytically derived physics-aware labels. Given the learned causal structure from the causal module, a forecasting module then integrates the learned representations with corresponding *causes* to predict *effects*. Our main contributions are summarized below:

- **Soft integration of physics equations via causality:** We propose a novel framework for the soft integration of physics laws. Causal relations are utilized as a soft encoding of the prior physics knowledge, without assuming access to precise underlying principles.

- **Physics-aware causal detection and retrieval:** We propose a semi-supervised causal structure learning using physics-aware causality. A causal module learns causal relations given additional explicit labels extracted from physics knowledge.

- **Superior empirical performance:** Through extensive evaluations, we show that our physics-aware causal model is robust in detecting similar patterns (i.e. inter-causality classification) and retrieving unlabeled causality (i.e. intra-causality retrieval) even with noisy and limited data. Our model P-STCGN improves the forecasting performance over real-word graph signals. Moreover, it excels in terms of data efficiency and generalization.

## 2  PROBLEM FORMULATION

We assume that a (static) graph structure $\mathcal{G}_s = (\mathcal{V}_s, \mathcal{E}_s)$ shared across different timestamps is given (or can be constructed by features of each node), along with observational data $\boldsymbol{X}_1, \cdots, \boldsymbol{X}_T$, where

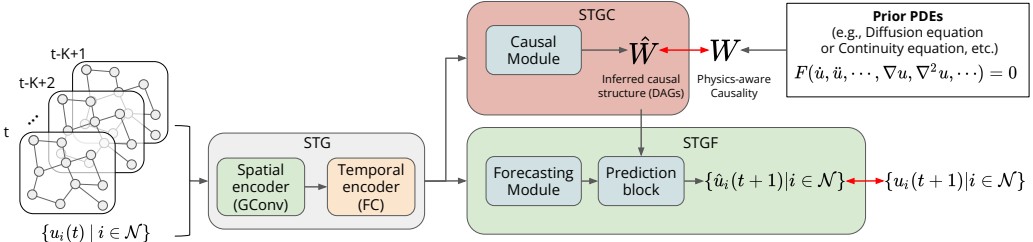

Figure 2: An overview of the proposed physics-aware spatiotemporal causal graph network (P-STCGN). A sequence of graph signals is firstly fed into a spatiotemporal graph network (STG). This is followed by two subsequent modules: (1) spatiotemporal graph networks for causality (STGC), and (2) spatiotemporal graph networks for forecasting (STGF). Prior PDEs from physical principles provide physics-aware causality. The red arrows denote how the supervised objectives are defined.

$\boldsymbol{X}_t \in \mathbb{R}^N$ are defined on the nodes in the graph. As there are $N$ different nodes, the observations ($\boldsymbol{X} \in \mathbb{R}^{T \times N}$) can be regarded as multivariate time series.

Additionally, we assume the existence of prior knowledge that could be moderately beneficial for modeling the observations. This is a mild assumption since real-world observations are usually governed by physical principles, such as meteorological measurements obtained from weather sensors in an Automatic weather station (AWS). One particularly important prior equation for turbulent dynamics is the Navier-Stokes equation (Wang et al., 2020a). These equations can be commonly represented as a function of spatial and time derivatives $F(\dot{u}, \ddot{u}, \cdots, \nabla u, \nabla^2 u, \cdots) = 0$, where $\dot{u}$ and $\ddot{u}$ denote the first and second-order time derivatives, respectively, and $\nabla$ represents the operator for spatial derivative. As the continuous operators can be numerically decomposed in a discrete domain (e.g. onto graph structures), we can explicitly define *causes* for a target observation at time $t$ and extract causal relations accordingly. Note that causal relations derived from a particular equation are only partially complete due to the uncertainty surrounding the true governing equation. The available prior knowledge need only be partially relevant to the underlying dynamics in order to be beneficial. The prior knowledge is primarily about structural dependencies, without assuming any access to associated parameters.

Given the physics-aware causal relations, we can assign explicit labels between $NK$ variables, where $K$ is a maximum time lag for causality. In the length $K$ observations $\boldsymbol{X}_{t-K+1}, \cdots, \boldsymbol{X}_t$, there are $NK$ total mutually correlated observations, and we define $N_c$ causal relations among the $NK \times NK$ possible relations. In Fig. 1, we have $N = 5$ nodes in $\mathcal{G}_s$, and the total number of variables in the length $K = 2$ sequence is 10. Thus, there are 100 possible relations between the 10 variables, and the Heat equation (Eq. 1) elucidates $N_c = 13$ (5 temporal and 8 spatial) causal relations. We denote the causal graph $\mathcal{G}_c = (\mathcal{V}_c, \mathcal{E}_c)$ where $|\mathcal{V}_c| = NK$ and $|\mathcal{E}_c| = N_c$.

Once the spatiotemporal data ($\boldsymbol{X}$) and the (partially available) causal relations ($\mathcal{G}_c$) are computed, our task is to find a model:

$$\hat{\boldsymbol{X}}_{t+1} = F(\boldsymbol{X}_{t-K+1}, \cdots, \boldsymbol{X}_t; \mathcal{G}_s, \mathcal{G}_c, \Theta), \tag{2}$$

where $\Theta$ is a set of learnable parameters in a model $F(\cdot)$.

## 3 PROPOSED MODEL

In this section, we describe the details of our proposed model, namely Physics-aware Spatiotemporal Causal Graph Networks (P-STCGN). The P-STCGN employs a two-stage learning approach to explicitly decouple causal structure learning from dynamic forecasting. Such an idea has been explored in (Löwe et al., 2022) for causal discovery. Fig. 2 shows a high-level view of P-STCGN consisting of two key modules: (1) spatiotemporal graph networks for causality (STGC), and (2) spatiotemporal graph networks for forecasting (STGF). STGC performs causal structure learning with causal labels derived from physics equations. Through STGC, we softly inject the inductive bias into our model. STGF then performs forecasting tasks using the learned causal structures. Both networks are designed to learn node representations from spatially and temporally correlated observations.

### 3.1 ARCHITECTURE

We first learn node-wise latent representations by two modules: spatial encoder (SE) and temporal encoder (TE). Spatial encoders are designed to learn spatial dependencies at each timestamp via the static graph structure $\mathcal{G}_s$. Graph convolutional networks such as GCN (Kipf & Welling, 2017) or GraphSAGE (Hamilton et al., 2017) are used to aggregate spatially neighboring information in a permutation invariant manner. The spatial encoder generates $K$ different snapshots which are grouped and fed into the temporal encoder as follows:

$$\{\boldsymbol{S}_{t'} = \text{SE}(\boldsymbol{X}_{t'}; \mathcal{G}_s) \mid t' = t - K + 1, \cdots, t\}, \tag{3}$$

$$\{\boldsymbol{Z}_{t'} = \text{TE}(\{\boldsymbol{S}_{t'-P}, \cdots \boldsymbol{S}_{t'}\}) \mid t' = t - K + 1, \cdots, t\}, \tag{4}$$

where $\boldsymbol{Z}_{t'} \in \mathbb{R}^{N \times D_c}$ is a set of node representations (dimension $D_c$) at time $t'$. $P$ is an aggregation order and TE merges a current embedding $\boldsymbol{S}_{t'}$ and past $P$ embeddings $\boldsymbol{S}_{t'-1}, \cdots, \boldsymbol{S}_{t'-P}$ for spatiotemporal node embeddings at $t'$. This temporal encoder does not consider the graph structure.

**Spatiotemporal graph networks for causality (STGC):** Once node embeddings are obtained, two $D_c$ dimensional vectors are fed into a causal module (CM), which computes a probability of causal association between the two corresponding nodes:

$$p_{ji}^{t_j t_i} = \text{CM}(\boldsymbol{Z}_{t_j,j}, \boldsymbol{Z}_{t_i,i}), \tag{5}$$

where CM is a fully connected network to compute causality and $\boldsymbol{Z}_{t_j,j}$ is the $j$-th node's representation at time $t_j$. Since there are $N$ different nodes at each time $t$ (with a total of $K$ different timestamps), there are $N^2 K^2$ different settings for $p$. Eq. 5 is similar to the *Key and Query* matching mechanism in the transformer (Vaswani et al., 2017). If observations are stationary and the causal relations are independent on the absolute timestamps $(t_j, t_i)$, but dependent on the relative time interval $\tau = t_i - t_j$, Eq. 5 can be reduced to $p_{ji}^\tau = \text{CM}(\boldsymbol{Z}_{t_j,j}, \boldsymbol{Z}_{t_i,i})$.

**Spatiotemporal graph networks for forecasting (STGF):** This module is used to learn node representations from spatiotemporal observations. It takes the learned causal structure from STGC and is used for the prediction of future signals. We introduce the forecasting module (FM) to transform the spatiotemporal representations ($\boldsymbol{Z}$) to task-specific representations. As CM learns causality-associated representations, FM is adapted to learn prediction-associated representations.

$$\{\boldsymbol{H}_{t'} = \text{FM}(\boldsymbol{Z}_{t'}) \mid t' = t - K + 1, \cdots, t\}, \tag{6}$$

where $\boldsymbol{H}_{t'} \in \mathbb{R}^{N \times D_v}$. Since the causal relations from the $NK$ past variables to $N$ variables are inferred from STGC, the causal probabilities $p_{ji}^{t_j t_i}$ (Eq. 5) are combined with $\boldsymbol{H}$ to predict next variables. Specifically, the output ($\boldsymbol{H}$) from FM in STGF and $p$ from CM in STGC are used to predict the next value at a node $i$ and time $t + 1$:

$$\hat{\boldsymbol{X}}_{t+1,i} = \sum_{t'=t-K+1}^{t-1} \sum_{j \in \mathcal{N}_i} p_{j,i}^{t't} \boldsymbol{H}_{t',j}. \tag{7}$$

It's worth noting that we use causal probabilities between $t' \in [t - K + 1, t - 1]$ and $t$ instead of $t' \in [t - K + 1, t]$ and $t + 1$. There are two reasons for this: (1) since $\boldsymbol{X}_{t+1}$ is not available, it is impossible to compute $p^{t',t+1}$ (a function of $\boldsymbol{X}_{t+1}$) in advance, and (2) we assume that the causality is stationary and thus from $t'$ and $t$ is invariant if $\tau = t - t'$ is unchanged. The second assumption is particularly valid for spatiotemporal observations in physical systems as most of physics-based phenomena are not dependent on the absolute time but relative time intervals.

### 3.2 TRAINING

**Additional causality labels from physics principles.** In Section 2, we assume that the causal relations are given as explicit labels based on the prior equation (Eq. 2). These causal relations are utilized for semi-supervised causal structure learning. This presents a few challenges when directly using these labels to update the causal module and the backbone, namely working around the selection bias of those labels provided by the physical priors.

The PDE solely provides information regarding which past and neighboring variables can be considered as possible *causes* for a current variable. It does not, however, provide information about which

causal relations should be *excluded*. Since the partially available labels are highly imbalanced, it is possible the CM will overfit on the positive-only labels. We address this challenge by introducing *non-causal* labels based on the principle that an effect can not occur before its cause. The non-causal labels are described as follows:

$$\{n_{ji}^{t_j t_i} = 0 \mid t_i - t_j < 0\}, \tag{8}$$

Eq. 8 captures the set of relations where a timestamp $(t_j)$ of a candidate cause $(\boldsymbol{X}_{t_j,j})$ is later than that of a candidate effect $(\boldsymbol{X}_{t_i,i})$. Despite the availability of the non-causal labels, the imbalance issue still exists as the cardinality of $\{n_{ji}^{t_j t_i}\}$ is much larger than that of $\{p_{ji}^{t_j t_i}\}$. We mitigate this by subsampling the non-causal labels as many times as the available causal labels.

## 4 EXPERIMENTAL RESULTS

We evaluate the proposed method in terms of both causal structure learning performance and dynamic forecasting performance. For causal structure learning, we evaluate the causal module (STGC) using synthetic and benchmark time series data. For dynamic forecasting performance, we evaluate P-STCGN through a graph signal prediction task with real-world observations.

### 4.1 CAUSAL STRUCTURE LEARNING EVALUATION

**Task formulation.** Given $N$ different stationary series (or nodes), we train a model to predict if there exists significant temporal causal relationships between two time series: $\boldsymbol{X}_{t',j}$ and $\boldsymbol{X}_{t,i}$. Since the auto-regressive order is $P$, there are potentially $NP \times N$ causal relations from $N$ variables $\boldsymbol{X}_{t'}$ where $t' \in [t - P, t - 1]$ to $N$ variables at time $t$. The true temporal causal relations are explicitly given as labels during training and a model is evaluated in two different aspects: (1) inter-causality classification and (2) intra-causality retrieval. For the *inter-causality classification*, we split a simulated multivariate time series into two parts across time axis: $\{\boldsymbol{X}_t | t = 1, \cdots, T_{train}\}$ and $\{\boldsymbol{X}_t | t = T_{train}, \cdots, T\}$. A model is trained from the first series (a training set) and evaluated in the second series (a testing set). For the *intra-causality retrieval*, we only use a subset of the known labels to train a model and evaluate the model if it can retrieve the unseen labels correctly.

**Baselines.** The task can be considered as *learning directional edge representations* from a variable at $t' \in [t - P, t - 1]$ to another variable at $t$, inspiring the three baselines as follows. First, we feed two node values into an MLP to predict the strength of causality. The other two baselines utilize a spatial and a temporal module to aggregate neighboring spatial/temporal values accordingly, after which the aggregated two node features are fed into an MLP to return the causal probability. For the spatial encoder (SE), we use GCN (Kipf & Welling, 2017), Chebyshev graph convolution networks (CHEB) (Defferrard et al., 2016), and GRAPHSAGE (Hamilton et al., 2017). The temporal encoder (TE) then concatenates node variables in the auto-regressive order. The STGC combines the two encoders spatiotemporally and the resultant node representations are fed into an MLP. Furthermore, we compare to causal discovery baselines: PARC (PCMCI (Runge et al., 2019b) based on partial correlations), Gaussian process regression and a distance correlation (GPDC), DYNOTEARS (Pamfil et al., 2020), and the SOTA Amortized Causal Discovery (ACD) (Löwe et al., 2022).

#### 4.1.1 SYNTHETIC STUDY

**Synthetic time series generation.** We first generate multivariate time series $\boldsymbol{X} \in \mathbb{R}^{T \times N}$ from known temporal causal relations. Consider $N$ different stationary time series where each series influences the others in a time-lagged manner. At time $t$, a variable in the $i$-th time series $\boldsymbol{X}_{t,i} \in \mathbb{R}$ is defined as a function of variables at $t' < t$ such that:

$$x_{i,t} = \sum_{t'=t-P}^{t-1} \sum_{j=1}^{N} f_{j,i}^{t',t}(\boldsymbol{X}_{t',j}) + \epsilon, \tag{9}$$

as described in [1] (Runge et al., 2019b), where $P$ is the auto-regressive order across time and $\epsilon$ is a noise term that is independent w.r.t. any other variable. Note that $f_{j,i}^{t',t}(\cdot)$ is regarded as a causal

---

[1]https://github.com/jakobrunge/tigramite

function from a previous variable at $(j, t')$ to a current variable $(i, t)$. Since the time series are stationary, the function $f_{j,i}^{t',t}(\cdot)$ in Eq. 10 can be relaxed as $f_{j,i}^{t-t'}(\cdot)$. We defined the temporal causal function in two different ways: (1) linear, and (2) non-linear conditional independence. For both settings, we generate length $T = 1000$ time series across $N = 7$ (linear) and $N = 13$ (non-linear) nodes. More detailed information can be found in the Appx. B.

| Model | Linear causality (AUC) | | Non-Linear causality (AUC) | |
|---|---|---|---|---|
| | $\mathcal{N}(0, 1^2)$ | $\mathcal{N}(0, 5^2)$ | $\mathcal{N}(0, 1^2)$ | $\mathcal{N}(0, 5^2)$ |
| MLP | 0.611±0.029 | 0.506±0.010 | 0.517±0.013 | 0.499±0.004 |
| GCN+MLP | 0.507±0.004 | 0.500±0.001 | 0.502±0.002 | 0.500±0.004 |
| CHEB+MLP | 0.627±0.010 | 0.513±0.008 | 0.526±0.009 | 0.500±0.004 |
| SAGE+MLP | 0.621±0.021 | 0.516±0.006 | 0.527±0.007 | 0.502±0.003 |
| TE+MLP | 0.827±0.021 | 0.697±0.012 | 0.562±0.033 | 0.511±0.009 |
| ACD | 0.476±0.031 | 0.489±0.020 | 0.495±0.013 | 0.504±0.010 |
| stGC(Ours) | **0.849±0.020** | **0.712±0.013** | **0.640±0.012** | **0.582±0.007** |

Table 1: Inter-causality classification with additional noise

**Inter-causality classification.** For the inter-causality classification task, we evaluate the proposed model on two different settings: (1) linear, and (2) non-linear temporal causality. The results on clean data (provided in Appx. C) demonstrate that the proposed model successfully outperforms other baselines on both settings. To evaluate the robustness of the proposed model, we intentionally add i.i.d. noises to the generated time series. Since the time series are "contaminated" by the random noise after being causally generated, it becomes much more difficult to discover underlying temporal causality. Table 1 shows AUC of the models on the linear and non-linear settings. While AUCs are commonly decreased, stGC can still learn meaningful representations from the spatiotemporal series unlike other methods. Note that when the scale of noise is increased ($\mathcal{N}(0, 5^2)$), MLP and spatial encoders followed by MLP are almost impossible to distinguish between causal and non-causal relations (AUC is closed to 0.5), occurring also to TE+MLP for the non-linear series.

Furthermore, in Table 1, the performance of ACD is poor which is likely due to the limited training data. ACD requires a large amount of training data to produce accurate causal classification (its default training size is 10,000). Besides, the additive noise is a significant bottleneck to the existing methods for temporal causal discovery in multivariate time series. Without the additional noise types, these methods are able to perfectly discover the causal directions. However, they significantly lose the capability to do so once the noise is included. The stGC, instead, can learn robust representations for effective causal discovery. Supportive results are discussed in the Appx. C where we compare stGC against PARC, GPDC, DYNOTEARS. Through comparisons, stGC is shown to be robust in causal classification with noisy and limited data by utilizing the physics-aware causality.

**Intra-causality retrieval.** For the intra-causality retrieval task, we used the time series generated from non-linear causality with added noises. Note that there are 21 causal relations in the series that are split into two parts for training and testing. By adjusting the number of causal relations shown in training series, we can evaluate how the proposed model is robust even if the majority of causal relations are not given during the training process. Table 2 shows the average behaviors trained on a subset of causal relations in time series. While TE+MLP detects some unseen causal relations when

| # of Train/Test causaliy labels | 16 / 5 | 11 / 10 | 6 / 15 | 1 / 20 |
|---|---|---|---|---|
| TE+MLP | 0.550±0.031 | 0.546±0.023 | 0.539±0.028 | 0.501±0.011 |
| ACD | 0.486±0.030 | 0.506±0.014 | 0.499±0.012 | 0.500±0.016 |
| stGC (Ours) | **0.636±0.024** | **0.620±0.010** | **0.596±0.014** | **0.585±0.018** |

Table 2: Intra-causality retrieval (AUC) from non-linear causal time series with $\mathcal{N}(0, 1^2)$

the number of labels shown for training is large (16), its performance quickly degrades as the number

of available labels is decreased. STGC outperforms TE+MLP by a large margin, supporting the claim that STGC can extract more informative spatiotemporal representations. Interestingly, even if only a single causal relation is given as a known label (1/20), STGC still manages to retrieve unseen causal relations. Due to the lack of sufficient training samples, the ACD fails to perform effective causal discovery. Compared to ACD, STGC, by leveraging the physics-aware causality, is able to retrieve unseen causal relations.

### 4.1.2 EVALUATION ON BENCHMARK DATASETS

**Three Benchmark Datasets** considered in the literature (e.g. (Löwe et al., 2022)) are employed: Particles, Kuramoto (Kuramoto, 1975), and Netsim (Smith et al., 2011). Particles and Kuramoto are two fully-observed physics simulations. Particularly, the Particles dataset simulates five moving particles in 2D. Some particles can influence others by pulling a spring. The Kuramoto dataset simulates five 1D time-series of phase-coupled oscillators. For both Particles and Kuramoto, we follow the same settings as the synthetic study and generate T=1000 time series for training. The Netsim dataset contains simulated fMRI data. The connectivity is defined between 15 brain regions. We follow the same settings as reported in (Löwe et al., 2022) and infer the connectivity across 50 samples.

| METHODS | PARTICLES | KURAMOTO | NETSIM |
|---|---|---|---|
| ACD | 0.493 | 0.562 | 0.688 |
| STGC (OURS) | **0.520** | **0.968** | **0.925** |

Table 3: Comparison to ACD on Benchmark Datasets.

Table 3 shows the results comparing STGC to ACD on the three benchmark datasets. From the results, we see that STGC significantly outperforms ACD. For example, on Kuramoto, P-STCGN achieves $40.6\%$ accuracy improvement compared to ACD. On Particles, the performance of ACD is poor which is likely due to the reduced data size. Though ACD achieves 0.999 AUC on the Particle dataset with sufficient data (10,000) as reported in (Löwe et al., 2022), its performance drops significantly with limited data (1,000). In contrast, we show that P-STCGN is able to learn robust representations for the causal discovery under limited data by utilizing the physics-aware causality.

### 4.2 DYNAMIC FORECASTING EVALUATION

To evaluate the dynamic forecasting performance of the proposed model, we consider a graph signal prediction task from real-world observations. The task is a prediction of future signals $X_{t+1}$ given length $P = 10$ past spatiotemporal series $X_{t-9} \cdots, X_t$ under the graph structure.

**Dataset:** We consider the climatology network[2] (Defferrard et al., 2020). Each sensor has 4 different daily measurements: TMAX: Maximum temperature (tenths of degrees C), TMIN: Minimum temperature (tenths of degrees C), SNOW: Snowfall (mm), and PRCP: Precipitation (tenths of mm). Each measurement is provided over 5 years from 2010 to 2014 (the length of series 1826), and we use them for our experiments. It is worth noting that the number of working sensors for each measurement is highly variable. While daily temperature observations are spatially densely available, the snowfall observations are comparatively sparse. We split the series into training (60%), validation (10%), and testing (30%) sets. Additional details are in Appx. B.

**Baselines:** We compare P-STCGN against two well-established data-driven baselines which have been introduced for similar tasks: DCRNN (Li et al., 2018) (Diffusion convolution recurrent neural network) and GCRN (Seo et al., 2018) (Graph convolutional recurrent network). For physics-based baseline, we consider the PA-DGN (Seo et al., 2019) (physics-aware difference graph network) and FNO (Li et al., 2020) (Fourier Neural Operator). FNO is not initially designed for forecasting given a history of data. We train FNO given observational data using its default settings and test it for a future

---

[2]Global Historical Climatology Network (GHCN) provided by National Oceanic and Atmospheric Administration (NOAA).

time step prediction. In addition to the external baselines, we evaluate against STGF, the version of the proposed model without the physics-aware causality. We exclude the comparison to existing data-driven large-scale simulations, such as Graph Network Simulator (GNS) (Sanchez-Gonzalez et al., 2020) and MeshGraphNets (Pfaff et al., 2020) since our primary focus is to incorporate physics for spatiotemporal modeling. Future work includes explore the proposed physics-aware pipeline for improving the data-driven large-scale simulation approaches.

**Causality labels from PDEs:** There is no ground truth PDEs for this dataset. We thus consider the PDEs among the family of the continuity equation, e.g. diffusion, convection, and Navier-Stokes equations. These equations commonly describe how target observations are spatiotemporally varying with respect to its second-order spatial derivatives and first-order time derivative. In the underlying graph structure, spatially 1-hop neighboring nodes ($j \in \mathcal{N}_i$) are considered as adjacent causes to the observation at the $i-$th node, and observations at $t - 1$ are potential causes to the observations at $t$ autoregressively. The existing causal labels can be described as $\{p_{ji}^{t_j t_i} = 1 \mid t_i - t_j = 1 \text{ and } j \in \mathcal{N}_i\}$.

### 4.2.1 PREDICTION ACCURACY

We use mean squared error (MSE) as a metric to compare P-STCGN against the external baselines[3]. Table 4 shows that P-STCGN mostly outperforms other baselines across different regions and measurements. Both DCRNN and GCRN replace fully connected layers in the RNN variants (GRU and LSTM) with diffusion convolution and Chebyshev convolution layers. Thus, they similarly aggregate spatiotemporal correlation, exemplified by the close prediction error. Compared to the data-driven DCRNN and GCRN, our approach achieves better accuracy, particularly for TMAX and PRCP. For example, P-STCGN improves DCRNN by $16\%$ for TMAX (western) prediction. Compared to two physics-based baselines PA-DGN and FNO, we also observe performance improvements. Particularly, P-STCGN improves PA-DGN significantly for TMAX and TMIN, demonstrating that our way of incorporating prior physics knowledge is much more effective.

| MEASUREMENT | TMAX | | TMIN | |
|---|---|---|---|---|
| MODEL | WESTERN | EASTERN | WESTERN | EASTERN |
| DCRNN | 0.1324±0.0024 | 0.1585±0.0033 | 0.0707±0.0017 | 0.1317±0.0028 |
| GCRN | 0.1336±0.0082 | 0.1588±0.0027 | **0.0701±0.0004** | 0.1302±0.0009 |
| FNO | 0.1234±0.0005 | 0.1963±0.0003 | 0.0906±0.0004 | 0.1676± 0.0001 |
| PA-DGN | 0.2620±0.0033 | 0.2921±0.0014 | 0.1720±0.0098 | 0.2346± 0.0009 |
| P-STCGN | **0.1111±0.0014** | **0.1355±0.0034** | 0.0731±0.0009 | **0.1262±0.0036** |

| MEASUREMENT | SNOW | | PRCP | |
|---|---|---|---|---|
| MODEL | WESTERN | EASTERN | WESTERN | EASTERN |
| DCRNN | 0.6757±0.0011 | 0.0406±0.0002 | 0.4703±0.0020 | 0.7588±0.0013 |
| GCRN | 0.6683±0.0012 | 0.0406±0.0001 | 0.4703±0.0009 | 0.7595±0.0001 |
| FNO | – | – | – | – |
| PA-DGN | 0.6626±0.0051 | 0.0402±0.0027 | 0.4979±0.0016 | 0.6819±0.0008 |
| P-STCGN | **0.6613±0.0035** | **0.0386±0.0007** | **0.4589±0.0033** | **0.6658±0.0025** |

Table 4: Summary of results of prediction error (MSE) with standard deviations.

### 4.2.2 ABLATION STUDY

To further study the effectiveness of P-STCGN, we perform ablation study on data efficiency and generalization ability. We compare to the baseline model STGF. Furthermore, we visualize the learned causality for interpretation.

**Data Efficiency.** We consider different training sets with reduced number of training samples: $\{5\%, 10\%, 20\%\}$. In comparison, we also consider the full training set (60%). All the testing are performed on the same testing split. Results are shown in Table 5. Compared to STGF, we can

---

[3]For SNOW and PRCP, FNO can't converge during training, likely due to the spatially sparse sensors with discrete measures.

clearly see how the additional physics-aware causality is beneficial for modeling spatiotemporal data, particularly on limited data (5%). Specifically, P-STCGN improves STGF by 28% on TMAX (western), implying that the PDE-based causal labels are significantly informative to compensate for the lack of training samples. As the training set size increases, the MSE difference between P-STCGN and STGF gets smaller. Nevertheless, P-STCGN continues to outperform STGF by a decent margin. With larger training set, the performance of both P-STCGN and STGF improves. Notably, for SNOW (eastern) and PRCP (western), there is a performance decrease as the training size increases from 20% to 40%. This decline is likely due to the noise in the training samples.

| | MODEL | 5% | 10% | 20% | 60% |
|---|---|---|---|---|---|
| TMAX(W) | STGF | $0.1926\pm0.0937$ | $0.1228\pm0.0014$ | $\mathbf{0.1177\pm0.0029}$ | $0.1134\pm0.0003$ |
| | P-STCGN | $\mathbf{0.1382\pm0.0034}$ | $\mathbf{0.1204\pm0.0005}$ | $0.1186\pm0.0007$ | $\mathbf{0.1111\pm0.0014}$ |
| TMAX(E) | STGF | $0.1611\pm0.0047$ | $0.1519\pm0.0039$ | $0.1410\pm0.0039$ | $0.1393\pm0.0011$ |
| | P-STCGN | $\mathbf{0.1584\pm0.0043}$ | $\mathbf{0.1493\pm0.0022}$ | $\mathbf{0.1404\pm0.0013}$ | $\mathbf{0.1355\pm0.0034}$ |
| TMIN(W) | STGF | $0.1229\pm0.0120$ | $\mathbf{0.0963\pm0.0063}$ | $0.0887\pm0.0040$ | $0.0759\pm0.0024$ |
| | P-STCGN | $\mathbf{0.1059\pm0.0080}$ | $0.0976\pm0.0012$ | $\mathbf{0.0874\pm0.0014}$ | $\mathbf{0.0731\pm0.0009}$ |
| TMIN(E) | STGF | $0.1571\pm0.0020$ | $0.1352\pm0.0112$ | $0.1263\pm0.0116$ | $0.1304\pm0.0038$ |
| | P-STCGN | $\mathbf{0.1427\pm0.0047}$ | $\mathbf{0.1283\pm0.0026}$ | $\mathbf{0.1232\pm0.0035}$ | $\mathbf{0.1262\pm0.0036}$ |
| SNOW(W) | STGF | $1.3300\pm0.0685$ | $0.9987\pm0.0100$ | $0.8150\pm0.0208$ | $0.6720\pm0.0070$ |
| | P-STCGN | $\mathbf{1.2223\pm0.0051}$ | $\mathbf{0.9783\pm0.0076}$ | $\mathbf{0.7977\pm0.0051}$ | $\mathbf{0.6613\pm0.0035}$ |
| SNOW(E) | STGF | $0.0460\pm0.0014$ | $\mathbf{0.0410\pm0.0003}$ | $0.0362\pm0.0003$ | $0.0391\pm0.0008$ |
| | P-STCGN | $\mathbf{0.0439\pm0.0005}$ | $0.0412\pm0.0083$ | $\mathbf{0.0356\pm0.0001}$ | $\mathbf{0.0386\pm0.0007}$ |
| PRCP(W) | STGF | $0.5103\pm0.0042$ | $0.4628\pm0.0020$ | $\mathbf{0.4407\pm0.0024}$ | $0.4619\pm0.0047$ |
| | P-STCGN | $\mathbf{0.5084\pm0.0012}$ | $\mathbf{0.4627\pm0.0014}$ | $0.4437\pm0.0022$ | $\mathbf{0.4589\pm0.0033}$ |
| PRCP(E) | STGF | $0.8028\pm0.0029$ | $0.8041\pm0.0060$ | $0.7980\pm0.0151$ | $0.6770\pm0.0042$ |
| | P-STCGN | $\mathbf{0.7982\pm0.0029}$ | $\mathbf{0.7981\pm0.0029}$ | $\mathbf{0.7884\pm0.0012}$ | $\mathbf{0.6658\pm0.0025}$ |

Table 5: Data efficiency evaluation. 60% represents the full training set. "W": Western. "E": Eastern.

**Generalization Ability.** We also consider the generalization ability whereby we train P-STCGN and STGF on one region and test on another region using the full training set. We observe that P-STCGN consistently outperforms STGF in most scenarios. The detailed results are reported in Appx. C.

**Interpretation of Learned Causality.** Once the causal module is trained based on a prior PDE, we can use it to examine how the potential causes are varying over space and time. We visualize how the causal probability is changed in Appx. C. From the visualizations, we can observe that variables spatially close to current observations have higher causal association for PRCP and TMAX. Additionally, we can infer the strength of causal relations between neighboring sensors and a specified target sensor (as shown in Appx. C). We find that the physics-aware causality is not only informative for spatiotemporal modeling but also enables the discovery of unspecified causal relations.

## 5   CONCLUSION

In this paper, we introduced a physics-aware spatiotemporal causal graph network (P-STCGN). Within our modeling process, we decoupled causal structure learning from dynamic forecasting. Causal relations were analytically derived from prior physics knowledge and used for semi-supervised causal learning. Subsequently, dynamic forecasting was performed based on the learned causal structure. We evaluated the proposed framework on time series data from two perspectives: causality classification and retrieval. Our evaluations demonstrated the effectiveness of the proposed physics-aware causal learning approach, especially in scenarios involving noisy and limited data. We further evaluated the forecasting performance on real-world observations from climate systems. Through evaluations, we demonstrated superior accuracy achieved through the utilization of physics-aware causality, without assuming the preciseness of the knowledge. In future work, we aim to explore the integration of alternative physics knowledge into P-STCGN and extend interpretability analyses to a broader range of real-world datasets.

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
