# A    RELATED WORK

**Physics-aware learning.** Physics-informed learning is an emerging research direction where physics knowledge as strong inductive biases is utilized in the construction of interpretable and robust deep models. Different approaches have been proposed to incorporate physics knowledge into deep models, such as model architecture design (de Bezenac et al., 2018; Seo et al., 2019). Employing physics principles as training loss is another widely adopted approach, such as the Hamiltonian neural network (Greydanus et al., 2019). Physics knowledge has been used for spatiotemporal modeling. Kaltenbach & Koutsourelakis (2021) propose a novel physics-aware generative state-space model for long-term predictions, and Wang et al. (2020a) marry two turbulent flow simulation techniques with deep neural networks to predict physical fields. Despite these achievements, physics-aware causality is not very well studied in this context.

**GNN-based Spatiotemporal Modeling.** While many graph-based architectures have been proposed to handle spatiotemporal observations on irregular domains (Li et al., 2018; Wu et al., 2020), one of the core issue behind graph-based neural networks (GNNs) (Hamilton et al., 2017; Kipf & Welling, 2017; Battaglia et al., 2018; Zhu et al., 2020) is that they cannot distinguish node-wise differences. To address this limitation, attention mechanisms have been extended to GNNs (Veličković et al., 2018; Zhang et al., 2018; Zheng et al., 2020). A recent work (Serrano & Smith, 2019), however, shows that attention weights do not necessarily correspond to importance for an output, but are instead more likely to be noisy predictors. Hence, there is a need for incorporation of *prior knowledge* into the unspecified representations between two nodes for reliable learning. Physics-aware causality can be a direct solution to this need, being effective for graph neural networks.

**Causal discovery in time series.** Discovering underlying causal structure in time series data is a fundamental problem that remains actively studied today (Runge, 2018; Runge et al., 2019a; Nauta et al., 2019; Runge et al., 2019b). Learning causal associations from time series is also an emerging topic in deep learning community. Pamfil et al. (2020) introduce a smooth acyclicity constraint to multivariate time series inspired by (Zheng et al., 2018). Amortized causal discovery (ACD) considered an explicit seperation between causal structure learning and the downstream dynamic prediction task (Löwe et al., 2022). Although many works are capable of discovering unknown causal structures from observational data directly, they usually assume data sufficiency, i.e., sufficient samples are available for accurate causal discovery. Besides, the performance can be sensitive to data noise. In contrast, we consider a semi-supervised causal discovery approach. We leverage explicit causal relations inferred from domain knowledge as physics-aware causality for robust causal classification and retrieval under limited and noisy data.

# B    EXPERIMENTAL SETTINGS

## B.1    DATA GENERATION AND PREPROCESSING

### B.1.1    SYNTHETIC TIME SERIES DATA

**Data Generation:**    We first generate multivariate time series $\boldsymbol{X} \in \mathbb{R}^{T \times N}$ from known temporal causal relations. Consider $N$ different stationary time series where each series influences the others in a time-lagged manner. At time $t$, a variable in the $i$-th time series $\boldsymbol{X}_{t,i} \in \mathbb{R}$ is defined as a function of variables at $t' < t$ such that:

$$x_{i,t} = \sum_{t'=t-P}^{t-1} \sum_{j=1}^{N} f_{j,i}^{t',t}(\boldsymbol{X}_{t',j}) + \epsilon, \tag{10}$$

as described in [4] (Runge et al., 2019b). Further breaking down this formulation seen in Section 4.1, we generate two series based on linear and nonlinear causality:

---

[4]https://github.com/jakobrunge/tigramite

**Linear causality:**

$$X_{t,0} = 0.7X_{t-1,0} + \epsilon$$
$$X_{t,1} = 0.8X_{t-1,1} + 0.8X_{t-1,3} + \epsilon$$
$$X_{t,2} = 0.5X_{t-1,2} + 0.5X_{t-2,1} + 0.6X_{t-3,3} + \epsilon$$
$$X_{t,3} = 0.4X_{t-1,3} + \epsilon$$
$$X_{t,4} = 0.9X_{t-2,2} + 0.1X_{t-3,6} + \epsilon$$
$$X_{t,5} = 0.2X_{t-1,0} + 0.2X_{t-2,0} + 0.2X_{t-3,0} + \epsilon$$
$$X_{t,6} = \epsilon$$

where $\epsilon \sim \mathcal{N}(0,1)$. In the linear causal series, there are 12 causal relations between $N = 7$ series and the maximum time lag in the causal relations is 3.

**Nonlinear causality:**

$$X_{t,0} = \epsilon$$
$$X_{t,1} = 0.2(X_{t-1,1})^2 + 0.7X_{t-2,2} + \epsilon$$
$$X_{t,2} = 0.3(X_{t-2,0})^3 + 0.05X_{t-1,3} + \epsilon$$
$$X_{t,3} = -0.09(X_{t-3,2})^2 + 0.4X_{t-1,5} + \epsilon$$
$$X_{t,4} = 0.2(X_{t-1,0})^2 + 0.01X_{t-3,1} - 0.2(X_{t-1,5})^2 + \epsilon$$
$$X_{t,5} = \epsilon$$
$$X_{t,6} = 0.3X_{t-1,5} + 0.3X_{t-2,4} - 0.3X_{t-3,3} + \epsilon$$
$$X_{t,7} = -0.2(X_{t-1,0})^2 + 0.7X_{t-2,8} + \epsilon$$
$$X_{t,8} = -0.3(X_{t-1,0})^3 + 0.05X_{t-2,0} + \epsilon$$
$$X_{t,9} = 0.9X_{t-3,1} + \epsilon$$
$$X_{t,10} = -0.02X_{t-1,0} + 0.1X_{t-3,6} - 0.2(X_{t-1,4})^2 + \epsilon$$
$$X_{t,11} = -0.3X_{t-4,0} + \epsilon$$
$$X_{t,12} = -0.3X_{t-1,11} + \epsilon$$

where $\epsilon \sim \mathcal{N}(0,1)$. In the nonlinear causal series, there are 22 causal relations between $N = 13$ series and the maximum time lag in the causal relations is 4 (See $X_{t,11}$). When we conduct the intra-causality retrieval experiment, we feed length 4 series from $X_{t-3}$ to $X_t$ to the classifier. Thus, the causality from time lag 4 in $X_{t,11}$ is not labelled. Figure 3 shows generated sample time series based on the formulation above.

**Data Preprocessing:**   Given $N$ different generated time series (length $T = 1000$), we assume a fully connected graph structure between the multivariate time series. We use the first 50% of the series data for the training series, and the following 20% for the validation series. The remaining 30% of the series is used to evaluate the baselines and P-STCGN.

### B.1.2   GRAPH SIGNAL DATA

**Data Preprocessing:**   We first sample time series from the entire sensor array to construct more localized graph signals. The number of available sensors is dependent on the type of measurement. Given multivariate time series from multiple sensors, we construct a distance-based graph structure using a $k-$NN algorithm where $k = 2$. The value of $k$ is chosen such that graph density is properly balanced, and to ensure that a sensor is only connected with other spatially close sensors. It is worth noting that the number of working sensors for each measurement is highly variable. While daily temperature observations are spatially densely available, the snowfall observations are comparatively sparse. Table 6 provides additional details for the dataset. The number of sensors from which the underlying graph was constructed is listed (along with the number of edges in the resulting graph).

Each observation is multiplied by a scalar (0.01) to be normalized and provide numerically stable computation. We used the first 60% of the series data for the training set, and the following 10% for the validation series. The remaining 30% series is used to evaluate the baselines and P-STCGN.

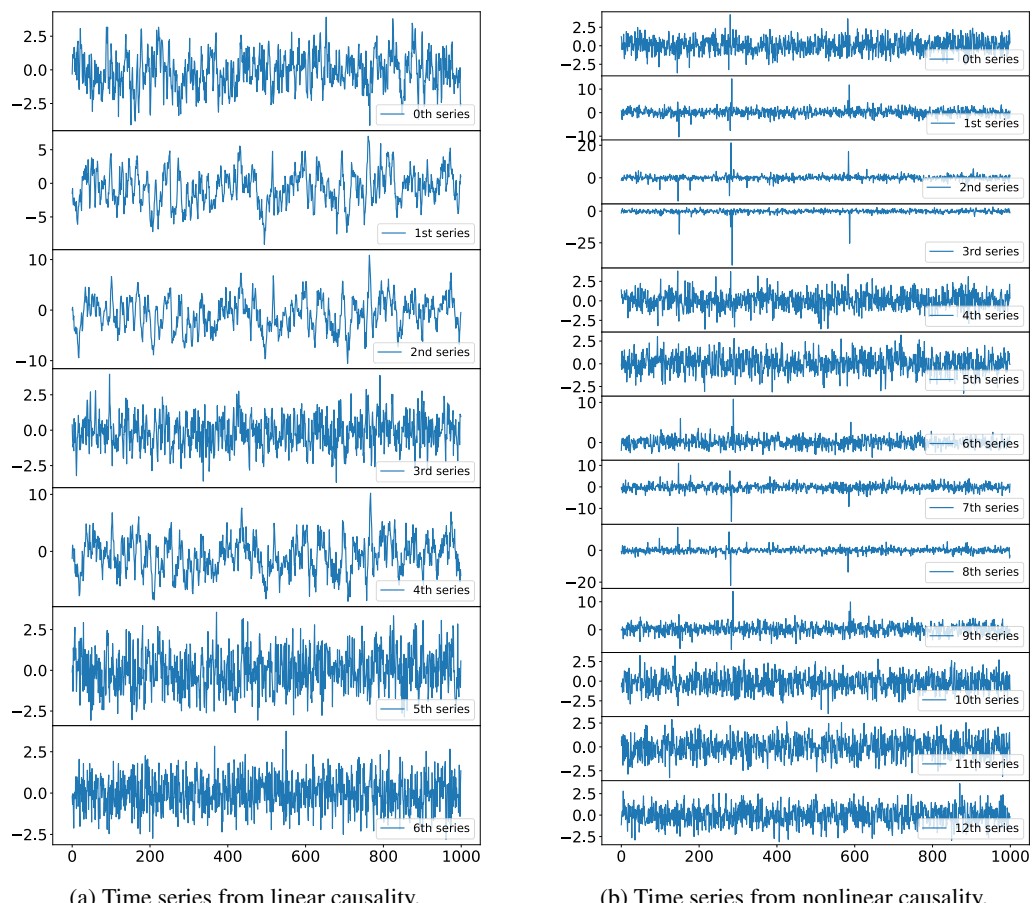

(a) Time series from linear causality.    (b) Time series from nonlinear causality.

Figure 3: Generated multivariate time series from given causal relations.

| WESTERN | TMAX | TMIN | SNOW | PRCP |
|---|---|---|---|---|
| # OF SENSORS | 434 | 423 | 31 | 319 |
| # OF EDGES | 1142 | 1110 | 76 | 862 |
| EASTERN | TMAX | TMIN | SNOW | PRCP |
| # OF SENSORS | 244 | 248 | 114 | 323 |
| # OF EDGES | 632 | 636 | 298 | 844 |

Table 6: Information on sensor networks from the climate dataset.

## B.2 MODEL CONFIGURATION

STG consists of two parts: SE and TE, and STG is shared in STGC and STGF. SE is defined by two-layer of GraphSAGE with 32 hidden units. TE concatenates the output of SE in a temporal axis, $[X_{t-P}; \cdots ; X_t]$. CM is MLP [FC32,ReLU,FC1,Sigmoid] and VM is MLP [FC32,ReLU,FC32,ReLU,FC1] where FC($n$) denotes a fully-connected layer with $n$ units. The baselines are defined to have similar number of learnable parameters from P-STCGN.

## B.3 TRAINING SETTINGS

We train P-STCGN for all tasks with a batch size of 32 on a single GPU (NVIDIA T4 GPU) for 1000 epochs with early stopping where a validation error is not improved for 20 epochs. All results in the paper are mean values from 10 different random seeds.

## C ADDITIONAL EVALUATIONS

### C.1 ABLATION STUDY ON CLEAN SYNTHETIC DATA

For the inter-causality classification task, we report the mean of recall, AUC, and cross entropy error (with standard deviation) on the test series. In this task, the temporal causality among the potential relations ($NP \times N$) is sparse so that the recall, which tells how many actual causal relations are retrieved, is particularly important. We evaluate the proposed model on two different settings: (1) linear, and (2) non-linear temporal causality. The results in Table 7 demonstrate that the proposed model successfully outperforms other baselines on both settings. More specifically, all models are able to distinguish non-causal and causal relations in the linear setting according to AUC. However, the temporal change is particularly important to understand the causality among the variables. For the non-linear setting, the results show that all metrics from models are degraded significantly compared to the linear setting. Nonetheless, the temporal information is more important but the spatial information can still be helpful (STGC vs. TE+MLP).

| MODEL | LINEAR CAUSALITY | | |
| --- | --- | --- | --- |
| | RECALL | AUC | CE |
| MLP | 0.579±0.124 | 0.670±0.012 | 0.611±0.015 |
| GCN+MLP | 0.193±0.126 | 0.508±0.008 | 0.669±0.004 |
| CHEB+MLP | 0.577±0.055 | 0.677±0.010 | 0.585±0.017 |
| SAGE+MLP | 0.554±0.161 | 0.668±0.035 | 0.583±0.014 |
| TE+MLP | 0.756±0.038 | 0.858±0.020 | 0.435±0.026 |
| STGC | **0.767±0.023** | **0.885±0.011** | **0.340±0.035** |
| MODEL | NON-LINEAR CAUSALITY | | |
| | RECALL | AUC | CE |
| MLP | 0.365±0.211 | 0.533±0.023 | 0.658±0.013 |
| GCN+MLP | 0.241±0.194 | 0.511±0.002 | 0.677±0.013 |
| CHEB+MLP | 0.416±0.124 | 0.551±0.013 | 0.650±0.011 |
| SAGE+MLP | 0.367±0.101 | 0.554±0.006 | 0.637±0.015 |
| TE+MLP | 0.438±0.107 | 0.611±0.051 | 0.625±0.017 |
| STGC | **0.503±0.041** | **0.689±0.013** | **0.522±0.015** |

Table 7: Inter-causality classification

### C.2 COMPARISON TO CAUSAL DISCOVERY METHODS

Table 8 shows recall from existing causal discovery in multivariate time series methods (PCMCI based on partial correlations (PARC) and Gaussian process regression and a distance correlation (GPDC)) on the non-linear series. It shows that STGC is able to learn robust representations for the causal discovery from noisy series by utilizing the explicitly given labels.

| NOISE | PARC | GPDC | DYNOTEARS | STGC |
| --- | --- | --- | --- | --- |
| $\mathcal{N}(0, 1^2)$ | 0.48 | 0.48 | 0.29 | 0.66 |
| $\mathcal{N}(0, 5^2)$ | 0.00 | 0.00 | 0.00 | 0.48 |

Table 8: Recall for causal discovery methods

### C.3 ABLATION ABILITY ON GRAPH SIGNAL PREDICTION

**Generalization ability.** To further study the effectiveness of incorporating the physics-aware causality, we study the generalization ability of P-STCGN. In particular, we perform an ablation study where we train P-STCGN and STGF on one region and test on another region. We consider the TMAX and TMIN for evaluation and the results are reported in Table 9.

| TMAX | WESTERN | | EASTERN | |
|---|---|---|---|---|
| | TRAIN ON WESTERN | TRAIN ON EASTERN | TRAIN ON EASTERN | TRAIN ON WESTERN |
| STGF | 0.1134±0.0014 | 0.1256±0.0027 | 0.1393±0.0011 | 0.1556±0.0024 |
| P-STCGN | **0.1111±0.0014** | **0.1240±0.0015** | **0.1355±0.0034** | **0.1532±0.0019** |
| TMIN | WESTERN | | EASTERN | |
| | TRAIN ON WESTERN | TRAIN ON EASTERN | TRAIN ON EASTERN | TRAIN ON WESTERN |
| STGF | 0.0759±0.0024 | **0.0906±0.0040** | 0.1304±0.0038 | 0.1308±0.0028 |
| P-STCGN | **0.0731±0.0009** | 0.0919±0.0021 | **0.1262±0.0036** | **0.1284±0.0014** |

Table 9: Generalization evaluation across the two regions.

### C.4 INTERPRETATION OF LEARNED CAUSALITY

Once the causal module is trained based on a guiding PDE, we can use the module to examine how the potential causes are varying over space and time. Fig. 4 shows how the causal probability is changed on the two regions. P-STCGN extracts causality-associated information from spatiotemporal series. In Fig. 4a, we can see that variables spatially close to current observations have higher causal assocation for PRCP and TMAX. A similar pattern appears on another region (shown in Fig. 4b). On the other hand, sensors for SNOW are more related to sensors farther away.

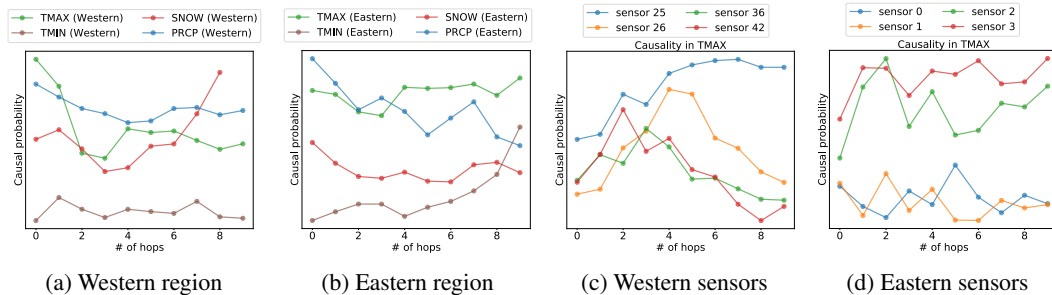

| (a) Western region | (b) Eastern region | (c) Western sensors | (d) Eastern sensors |

Figure 4: (a,b) Average causal probability curves vs. the number of hops over all sensors in each region. (c,d) Average causal probability curves vs. the number of hops from particular sensors in each region (TMAX)

We can also infer which neighboring sensors have stronger/weaker causal relations to a specified target sensor. In Fig. 4c and 4d, 4 sensors are sampled from each region to visualize how much their $K-$hop neighboring variables are causally related. We can see that daily max temperatures from the sensor 42 in the western region have been strongly affected by spatially close (2-hop) sensors, however, the max temperature at the sensor 25 is more likely dependent on sensors a bit far away (6 or 7-hop). On the other hand, sensor 26 is more dependent on mid-range sensors (4 or 5-hop). In eastern states, sensors 2 and 3 are associated with closer sensors; however, sensor 0 and 1 do not have distinct causal relations from their neighboring sensors. We find that the physics-aware causality is not only informative for spatiotemporal modeling directly but also enables the discovery of unspecified causal relations.