# OpenReview forum: "Physics-aware Causal Graph Network for Spatiotemporal Modeling"
_ICLR.cc/2024/Conference — Submitted to ICLR 2024_

### Official Review · Reviewer_EyEy · 2023-10-15

**Soundness:** 3 good
**Presentation:** 3 good
**Contribution:** 2 fair
**Rating:** 5
**Confidence:** 4

**Summary:**

The paper proposes an approach for spatiotemporal modeling using physics-aware causality. The approach integrates domain knowledge with data-driven models to construct more robust and interpretable pipelines. The model learns causal weights from spatially close and temporally past observations to current observations via semi-supervised learning, allowing it to capture the underlying cause-effect relationships in the data. The approach employs a regularization term to capture the causal structures that align with the physics-aware causality, further improving the model's performance. The model's ability to handle noisy and limited data is demonstrated by extensive experiments on time series data, and its superior forecasting performance on real-world graph signals highlights its effectiveness in capturing the underlying physics principles governing spatiotemporal observations. Overall, the proposed approach is a promising direction forward in the field of machine learning, with potential applications in climate, traffic systems, and electricity networks.

**Strengths:**

1. The manuscript is clearly written, with a well-defined motivation. Incorporating causality into spatio-temporal data mining presents an intriguing perspective.

2. The experiments conducted are reasonable, validating the model's capabilities across multiple scenarios.

**Weaknesses:**

1. In real-world scenarios, a plethora of spatio-temporal graph data, often comes with inherent noise from various sources. Typically, methods based on physical principles might not perform well in real-world settings and can exhibit weak generalization capabilities. I would like to see the authors conduct tests on real-world graph data to better demonstrate the generalization ability of their model.

2. The authors should consider testing their approach on more challenging datasets, such as the n-body system, to further validate its effectiveness.

3. While the paper emphasizes causal theory, I did not observe explicit references or applications of foundational causal concepts, such as backdoor adjustment or front door adjustment. I encourage the authors to elucidate the underlying causal motivations and proofs in greater detail. This would significantly strengthen the paper's results and its overall contributions.

**Questions:**

see weakness.

---

> ### Author Response · Authors · 2023-11-22
> **Responses to Reviewer EyEy**
>
> Thank you for the encouraging comments and the helpful suggestions. We answer the questions below.
>
> **Q1 [Generalization Evaluation]:** We conducted the generalization evaluation on real-world graph data as shown in Appendix G.3. In particular, we perform an ablation study where we train P-STCGN on one region and test on another region for climate forecasting.  Table 9 shows that our proposed model achieves improved generalization performance compared to the baseline.
>
> **Q2 [Evaluation on more challenging tasks]:** Thank you for the suggestion. We will consider the more challenging n-body system for further evaluation.
>
> **Q3 [Clarity on foundational causal concepts]:** While our paper doesn’t aim to develop a comprehensive causal discovery theory, we appreciate your suggestion to enhance clarity on foundational concepts. In the current presentation, we primarily focus on motivating the causal representation of physics equations as discussed in the Introduction and Problem Formulation sections. We provide the detailed discussions on causal discovery in time series in Appendix A.

---

### Official Review · Reviewer_cRbd · 2023-10-22

**Soundness:** 3 good
**Presentation:** 2 fair
**Contribution:** 3 good
**Rating:** 3
**Confidence:** 4

**Summary:**

This paper introduces the Physics-aware Spatiotemporal Causal Graph Network (P-STCGN), a approach that softly integrates the laws of physics into causal graph structures. This integration aims to enhance the robustness of spatiotemporal models by leveraging valuable inductive biases from interpretable physics equations. The primary challenges addressed involve the mismatch between the assumptions of existing models and real-world observations. The P-STCGN model capitalizes on the inherent causal relationships present in physical dynamics across both space and time.

**Strengths:**

(1) The authors' consideration of introducing causality into model construction is quite intriguing. Moreover, this approach contributes significantly to the model's interpretability.

(2) The authors adeptly address the potential challenges of capturing real-world physical laws, showcasing a strong foundation in practical scenarios.

**Weaknesses:**

(1)	The coherence between the introduction and the main content of the paper is somewhat lacking. The introduction mentions the existence of various physical laws in the real world. However, in the model and specific dataset experiments, these physical laws were not introduced as prior knowledge. I am curious whether this prior was learned or predefined. Moreover, since the introduction (in the second paragraph) states that partial physical laws can be discovered from real-world data, I wonder why no tests on related real data were conducted to verify this claim. These inconsistencies negatively impact the overall coherence of the paper, making it challenging to read.

(2) The paper emphasizes modeling using causal theory. However, learning causal correlations through an MLP (Multi-layer Perceptron) seems somewhat inappropriate. In other words, the causal relationships in the model are determined by a black-box network. This approach might require further justification. While I completely understand the use of attention mechanisms to establish interpretability and correlation, an MLP seems to lack a logical foundation in this context.

**Questions:**

(1) The authors should consider comparing their approach with some state-of-the-art spatio-temporal GNN frameworks in the experiments. This would provide a clearer context for the performance and relevance of the proposed method.

(2) The integration of causal relationships with real-world equations should be further elucidated in the model description. It would be beneficial for readers to have a comprehensive understanding of how these components interact within the proposed framework.

---

> ### Author Response · Authors · 2023-11-22
> **Responses to Reviewer cRbd**
>
> Thank you for the encouraging comments and the helpful suggestions. We answer the questions below.
>
> **Q1 [Comparisons to more baselines]:** While we primarily focus on comparison to physics-informed frameworks, we appreciate the suggestion and will consider more SOTA data-driven frameworks.
>
> **Q2 [Clarifications on the prior]:** In our real-world dataset evaluations, we described the causal labels from PDEs. Since there is no ground truth PDEs for the dataset, we consider PDEs among the family of the continuity equation, e.g., diffusion, convection, and Navier-Stokes equations. The prior is pre-defined. We will further clarify this in the revision.
>
> **Q3 [Discovering partial physical laws]:** We apologize for the confusion. The primary goal of this paper is not to perform equation discovery. We are aimed at introducing a soft integration of physics laws, that are potentially incomplete and noisy, into deep models through causal learning. By doing this, our approach doesn’t rely on the assumption that the precise prior knowledge is accessible. We will clarify this statement in our introduction. We value the reviewer’s point, and we find it very interesting. We would like to study the effectiveness of causal approaches in helping to discover partial physics laws as our future work.

---

### Official Review · Reviewer_B5jF · 2023-10-31

**Soundness:** 3 good
**Presentation:** 2 fair
**Contribution:** 3 good
**Rating:** 6
**Confidence:** 4

**Summary:**

This paper introduces a novel approach called Physics-Aware Spatiotemporal Causal Graph Network (P-STCGN) for integrating physical equations into spatiotemporal models. The idea is to leverage causality to capture the fundamental causal relations present in physics dynamics. The proposed approach uses a causal module to learn causal weights from past observations to current observations and a forecasting module to perform predictions guided by cause-effect relations. Evaluations conducted on synthetic as well as real-world climate datasets demonstrate the superior performance for the proposed method.

**Strengths:**

1. The Integration of Physics Knowledge is quite innovative.
2. The paper provides an extensive evaluation of the proposed method on different datasets.

**Weaknesses:**

Weaknesses/questions
1.	How does the model perform when the prior physics knowledge is ambiguous or not well established? How to verify the accuracy of the physics knowledge being integrated?
2.	Can the authors elaborate on why the model is able to handle the noisy data?
3.	Besides the climate-related application, how easy it is to extend the model to other domains?

**Questions:**

See weaknesses above.

---

> ### Author Response · Authors · 2023-11-22
> **Responses to Reviewer B5jF**
>
> Thank you for the encouraging comments and insights. We answer the questions below.
>
> **Q1 [Ambiguous prior knowledge and noisy data]:** The goal of the paper is to address the challenge due to limited access to precise knowledge, which can be caused by the ambiguity of the prior knowledge or data noise. Our assumption is that the prior knowledge is moderately beneficial for modeling observations. In our causal structure learning evaluations, we show that our approach is robust to data noise by leveraging prior physics knowledge. In our real-world evaluation where ground truth PDEs are not available, we adopt the family of the continuity equation. Through the proposed semi-supervised causal structure learning, our model still achieves improved performance.
>
> **Q2 [Verify the accuracy of integrated prior knowledge]:** We verify the accuracy through the extensive causal structure learning evaluation using both synthetic data and benchmarks. In these settings, we know the ground truth equations for generating the data and thus we quantitatively measure the accuracy of the learned caudal structures with prior knowledge integrated.  Results demonstrate that our approach can perform accurate causal structure discovery, aligning well with prior knowledge.
>
> **Q3 [Extending to other domains]:** Our approach is domain-agnostic. Once moderately beneficial physics equations are identified, our method can be straightforwardly applied to any domain.

---

### Official Review · Reviewer_pHbu · 2023-11-05

**Soundness:** 2 fair
**Presentation:** 3 good
**Contribution:** 2 fair
**Rating:** 3
**Confidence:** 5

**Summary:**

This paper aims to integrate a spatio-temporal graph neural network with physics-aware causality for spatio-temporal modeling. The major contribution is the soft integration of physics equations with causality. Experiments over several synthetic and real-world datasets can verify the effectiveness of the proposed model.

**Strengths:**

1. The paper is well-written and easy to follow. Integrating spatio-temporal graph neural network is of great importance to many real-world applications.
2. The paper conducts experiments over both synthetic and real-world datasets.

**Weaknesses:**

My major concerns are:
1. Insufficient related work. To the best of my knowledge, there is quite a large number of literature exploring the integration of physics law or causality into spatio-temporal graph neural networks [1,2,3,4,5,6]. For example, [1,2,6] employ neural ordinary differential equations to capture continuous ST dependencies. Ji et al. propose a physics-guided neural network for spatiotemporal modeling in traffic flows [3]. CaST designs a new framework for handling causality in spatio-temporal graphs [4]. However, this paper lacks a discussion on these studies and doesn't compare the proposed model with them either. What's the difference between them? Why should we use the proposed model? It would be good to survey more related publications before paper submission. What's more, the related work section should be included in the main body of the paper, instead of the appendix.
2. The technical contribution of this work against existing approaches is not significant, which is clearly below the acceptance level of ICLR.
3. The term "causality" in this paper is questionable. This causality is more similar to proximity in other spatio-temporal graph neural networks [7, 8], rather than the actual causality in causal inference.
4. The learned causality in this paper lacks justification.
5. The baselines used in this paper are weak and outdated. Please consider more recent baselines for comparison (see the above references).
6. This paper lacks the experiment over one of the most popular tasks -- traffic forecasting, which is also driven by inherent physics laws.
7. No source code for reproducing the results.
8. No discussion on the model efficiency and model size.

Reference:

[1] Fang, et al. "Spatial-temporal graph ode networks for traffic flow forecasting." SIGKDD 2021.

[2] Choi,et al. "Graph neural controlled differential equations for traffic forecasting." AAAI 2022.

[3] Ji et al. "STDEN: Towards physics-guided neural networks for traffic flow prediction." AAAI 2021.

[4] Xia et al. "Deciphering Spatio-Temporal Graph Forecasting: A Causal Lens and Treatment." NeurIPS 2023.

[5] Jia et al. "Physics-guided recurrent graph model for predicting flow and temperature in river networks." SDM 2021.

[6] Liang et al. "Mixed-order relation-aware recurrent neural networks for spatio-temporal forecasting." TKDE 2022.

[7] Wu et al. "Graph wavenet for deep spatial-temporal graph modeling." IJCAI 2019.

[8] Bai et al. "Adaptive graph convolutional recurrent network for traffic forecasting." NeurIPS 2020.

**Questions:**

Please reply to the questions in the weaknesses.

---

> ### Author Response · Authors · 2023-11-22
> **Responses to Reviewer pHbu**
>
> Thank you for the comments and suggestions. We reply to the questions below.
>
> **Q1 [Insufficient related work and baselines]:** Thank you for pointing out these references. We will check them and properly incorporate them into our revision.
>
> **Q2 [Technical novelty and the term ‘Causality’]:** The technical novelty of our approach lies in a systematically soft integration method, incorporating prior physics equations into deep models through causal structure learning. This approach is non-trivial, as the existing physics-informed deep learning literature had shown limited focus on the causal representation of physics equations. Our approach distinctly deviates from existing works [7,8], as our goal is to leverage prior physics equations, thereby relaxing assumptions on accessing precise physics knowledge.
>
> **Q3 [Justifications of causality]:** In section 4.1, we perform extensive evaluation on causal structure learning on synthetic and benchmark time series data. In these settings, we know the ground truth equations for generating the data and thus we quantitatively measure the accuracy of the learned caudal structures with prior knowledge integrated.  Results demonstrate that our approach can perform accurate causal structure discovery, aligning well with prior knowledge.
>
> **Q4 [Evaluations on traffic forecasting]:** Thank you for the suggestions. We will extend our approach to traffic forecasting to further evaluation its effectiveness as future work.

---

> > ### Comment · Reviewer_pHbu · 2023-11-22
> > **Response to authors**
> >
> > Thank you for replying to these questions. I hope the paper can be improved based on these comments.

---

### Meta-Review · Area_Chair_oCCD · 2023-12-07

**Metareview:**

The paper aims to combine concepts from causality and physics-informed ML. This is a very interesting and promising idea. However, the reviewers remarked that these ideas are not followed through, that it is unclear whether and how causal and physics-informed concepts are implemented, and criticized the experimental evaluation.

**Justification For Why Not Higher Score:**

The criticisms of the reviewers are rather explicit and clearly indicate a reject. It is unclear whether this work really uses causal and/or physics-informed concepts.

**Justification For Why Not Lower Score:**

NA

---

### Decision · Program_Chairs · 2024-01-16

Reject